# Capturing children food exposure using wearable cameras and deep learning

**Shady Elbassuoni**[1]*, **Hala Ghattas**[2,3], **Jalila El Ati**[4], **Yorgo Zoughby**[1], **Aline Semaan**[5], **Christelle Akl**[2], **Tarek Trabelsi**[4], **Reem Talhouk**[6], **Houda Ben Gharbia**[4], **Zoulfikar Shmayssani**[1], **Aya Mourad**[1], **with SCALE Research Group**[¶]

**1** Computer Science Department, American University of Beirut, Beirut, Lebanon, **2** Center for Research on Population and Health, American University of Beirut, Beirut, Lebanon, **3** Department of Health Promotion, Education, and Behavior, University of South Carolina, Columbia, South Carolina, United States of America, **4** SURVEN Research Laboratory, National Institute of Nutrition and Food Technology, Tunis, Tunisia, **5** Department of Public Health, Institute of Tropical Medicine, Antwerp, Belgium, **6** School of Design, Northumbria University, Newcastle upon Tyne, United Kingdom

¶ Membership of the SCALE Research Group is listed in the Acknowledgments.
* se58@aub.edu.lb

**Data Availability Statement:** The data is now uploaded on AUB ScholarWorks and can be requested through scholarworks@aub.edu.lb. Here are the links to the datasets on AUB ScholarWorks: Neighborhood Mapping:

## Abstract

Children's dietary habits are influenced by complex factors within their home, school and neighborhood environments. Identifying such influencers and assessing their effects is traditionally based on self-reported data which can be prone to recall bias. We developed a culturally acceptable machine-learning-based data-collection system to objectively capture school-children's exposure to food (including food items, food advertisements, and food outlets) in two urban Arab centers: Greater Beirut, in Lebanon, and Greater Tunis, in Tunisia. Our machine-learning-based system consists of 1) a wearable camera that captures continuous footage of children's environment during a typical school day, 2) a machine learning model that automatically identifies images related to food from the collected data and discards any other footage, 3) a second machine learning model that classifies food-related images into images that contain actual food items, images that contain food advertisements, and images that contain food outlets, and 4) a third machine learning model that classifies images that contain food items into two classes, corresponding to whether the food items are being consumed by the child wearing the camera or whether they are consumed by others. This manuscript reports on a user-centered design study to assess the acceptability of using wearable cameras to capture food exposure among school children in Greater Beirut and Greater Tunis. We then describe how we trained our first machine learning model to detect food exposure images using data collected from the Web and utilizing the latest trends in deep learning for computer vision. Next, we describe how we trained our other machine learning models to classify food-related images into their respective categories using a combination of public data and data acquired via crowdsourcing. Finally, we describe how the different components of our system were packed together and deployed in a real-world case study and we report on its performance.

http://hdl.handle.net/10938/23982 Children Trajectory: http://hdl.handle.net/10938/23981 Food Consumption Images: http://hdl.handle.net/10938/23980.

**Funding:** This work is supported by the International Development Research Centre (IDRC, http://www.idrc.ca/) in Canada, award number 108641103657 (S.E., H.G., J.E.A.). The funders had no role in study design, data collection and analysis, decision to publish, or preparation of the manuscript. All the authors received salaries from the grant.

**Competing interests:** The authors have declared that no competing interests exist.

## Author summary

Capturing food exposure of school children is a challenging task due to recall bias. In this manuscript, we describe a machine-learning-based data-collection tool that can automatically record school children's exposure to food items, food advertisements and food outlets in their homes, schools and neighborhoods. Our data-collection tool consists of a wearable camera to capture continuous footage of children's environments during a typical school day, and a series of machine learning models that can extract food-related images from the recorded footage and classify them into images that contain food items consumed by the child wearing the camera, or consumed by others, images that contain food advertisements, and images that contain food outlets. We report on a user-centered design study that assessed the acceptability of using wearable cameras to capture food exposure among school children in two urban Arab centers, namely Greater Beirut in Lebanon and Greater Tunis in Tunisia. We then describe how we trained our various machine learning models to capture food exposure among school children and categorize such food exposure into a predefined typology. Finally, we also report on the results of deploying our data-collection tool in a real-world case study in Tunisia.

## 1 Introduction

Children's food choice drivers are elicited in complex frameworks of interconnected factors at the levels of their homes, schools and neighborhoods. Evidence shows that children's exposure to food advertisements and shops in their daily environments, particularly on their trajectories to and from school is a main contributing element to their food choices and dietary behaviors [1–4]. Capturing these exposures' presence and measuring their frequency and potential associations with children's dietary habits and health and nutrition outcomes is commonly limited by the use of traditional methods of data collections that are subject to information and recall bias [5–7]. Technology-based tools enable an objective and comprehensive measurement of children's nutrition-related behaviors and experiences around food [6, 8, 9]. Digital technologies can engage participants as active contributors to the research process, objectively document their lived experiences, foster a "people-based" approach to measuring these exposures [10], with children being at the center of the process, which may also lead to more accurate and representative data on their food experiences [9, 11].

There is a wealth of research conducted on technology-based dietary assessment [12]. For example, Lui et al. [13] proposed a wearable-sensor platform that can automatically provide information regarding a subject's dietary habits. Similarly, Signal et al. [14] reported on innovative research in New Zealand in which children wore cameras to examine the frequency and nature of everyday exposure to food marketing across multiple media and settings. A follow-up study by McKerchar et al. [15] focused on food-store environments and assessed food-product availability, placement, packaging, branding, price promotion, purchases and consumption. Another follow-up study by Liu et al. [16] focused on space-time analysis of unhealthy food advertising among the school children using the data captured through the wearable cameras.

Gao et al. [17] explored the feasibility of applying Simultaneous Localization and Mapping (SLAM) techniques for continuous food volume measurement with a wearable monocular camera. Shroff et al. [18] proposed DiaWear, a novel assistive mobile-phone-based calorie-monitoring system to improve the quality of life of individuals with special nutrition management needs. Davies et al. [19] explored using wearable cameras to monitor eating and drinking

behavior during transport journeys. Similarly, Gage et al. [20] analyzed the frequency and context of snacking among children using wearable cameras. Doulah et al. [21] proposed a novel wearable sensor system that uses a combination of acceleration and temporalis muscle sensors for accurate detection of food intake and triggering of a wearable camera to capture the food being consumed.

Jia et al. [22] developed a machine-learning-based approach, which can automatically detect food items from images acquired by an egocentric wearable camera for dietary assessment. Similarly, Jia et al. [23] developed a small, chest-worn electronic device called eButton, which automatically took pictures of consumed food for objective dietary assessment. Finally, Hossain et al. [24] proposed a machine-learning-based system which classified captured images by wearable egocentric cameras as food or no-food images in real-time, which is the closest to our work.

Although many of the above-surveyed technologies have been positively received by younger participants, and are considered to be feasible and acceptable means of documenting food exposures [7–9, 25], they have not been used and validated in low-middle income countries or outside of a Western cultural context. Thus, the purpose of this manuscript is two-fold:

1- To describe a machine-learning-based data-collection system that captures children's food exposure using wearable cameras; this system is informed by (a) the literature and (b) a user-centered design study in order to ensure an ethical, secure and scalable system.

2- To validate the precision and accuracy of the proposed system in the real world using interim data from a case study conducted in Tunisia.

The settings of this work are two urban agglomerations in the Arab region; the Greater Beirut Area in Lebanon and Greater Tunis in Tunisia. These two middle-income countries have witnessed a rapidly-proceeding nutrition transition in the past two decades and the estimates of childhood overweight match those of high-income countries, reaching about 30% in Lebanon and 20% in Tunisia [26].

We developed a data collection system that uses wearable cameras to continuously capture videos of the child's environment during a typical school day. The captured footage acts as a recorded diary of what the child wearing the camera is being exposed to. Since the amount of footage captured is typically large, and only a small percentage corresponds to images with content related to food, we employ machine learning to identify images (i.e., video frames) related to food and discard any other footage. We consider that an image is related to food or is a food exposure image if it displays anything associated with food and/or beverages such as food items, food outlets, restaurants, food brands, food markets, food advertisements, and food vending machines, etc. We also employed machine learning to classify food exposure images into images that contain food items being consumed by the child wearing the camera or others, images that contain food advertisements, and images that contain food outlets. Fig 1 gives an overview of our food exposure typology and Fig 2 (a) shows example images that contain food consumption, (b) food outlets, (c) food advertisements, and (d) both food outlets and food advertisements.

In order to develop such a protocol, we first assessed the acceptability of using wearable cameras among school children through a user-centered design study involving school staff, parents and school children (10-12 years old), in 12 interactive design workshops conducted in Greater Beirut and Greater Tunis. These workshops employed discussions, mind mapping and storyboarding activities to identify challenges associated with the use of wearable cameras among school children participating in research, and inform the design of a tool that meets cultural and ethical requirements. Discussions during the workshops were audio-recorded, transcribed and analyzed thematically along with the mind maps and storyboards.

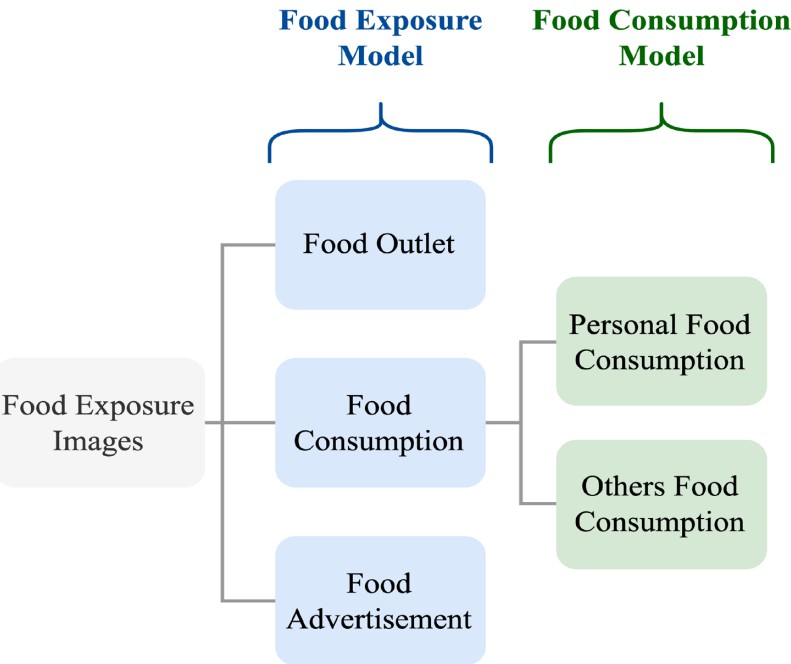

**Fig 1. Overview of the food exposure typology.**

Commonly reported challenges were: invading children's and third-parties' privacy, distracting school children during classes, and obtaining biased data. To overcome these challenges, participants suggested wearable cameras capturing exposure automatically for a short period of time. Two rounds of image filtering were proposed to safeguard privacy: automated selection of images related to food exposure, followed by parental manual screening. To protect anonymity, participants suggested automated blurring of faces in all captured footage.

Building on the ethical and cultural requirements that were identified in our user-centered design study, we developed a machine-learning-based data-collection system that automatically captures images with content related to food using wearable cameras. The system includes:

1- A wearable camera that satisfies the needed requirements.

2- A machine learning model for automatic detection of images with content related to food. To train such a model, we built a dataset consisting of images that were crawled from the Web using Google and Microsoft Bing image search services. Our dataset was divided into two equally sized classes: food exposure and non-food exposure. The first class consists of images containing food items, food outlets, restaurants, food brands, food ads, food markets, etc. The non-food exposure class consists of objects/places unrelated to food and commonly seen during a typical school day (e.g., classroom, playground, interiors of buses, cars, books, TV screens, bedrooms, living rooms, trees, mountains, sea, and animals). The images in our dataset were automatically labeled based on the search query that returned each image. We then used this dataset to train a Mobile-Net V1 convolutional neural network model [27] that achieved an *average F1-score of 0.92* on test data.

3- A face blurring technique to blur faces on the set of images retained after deleting images not related to food.

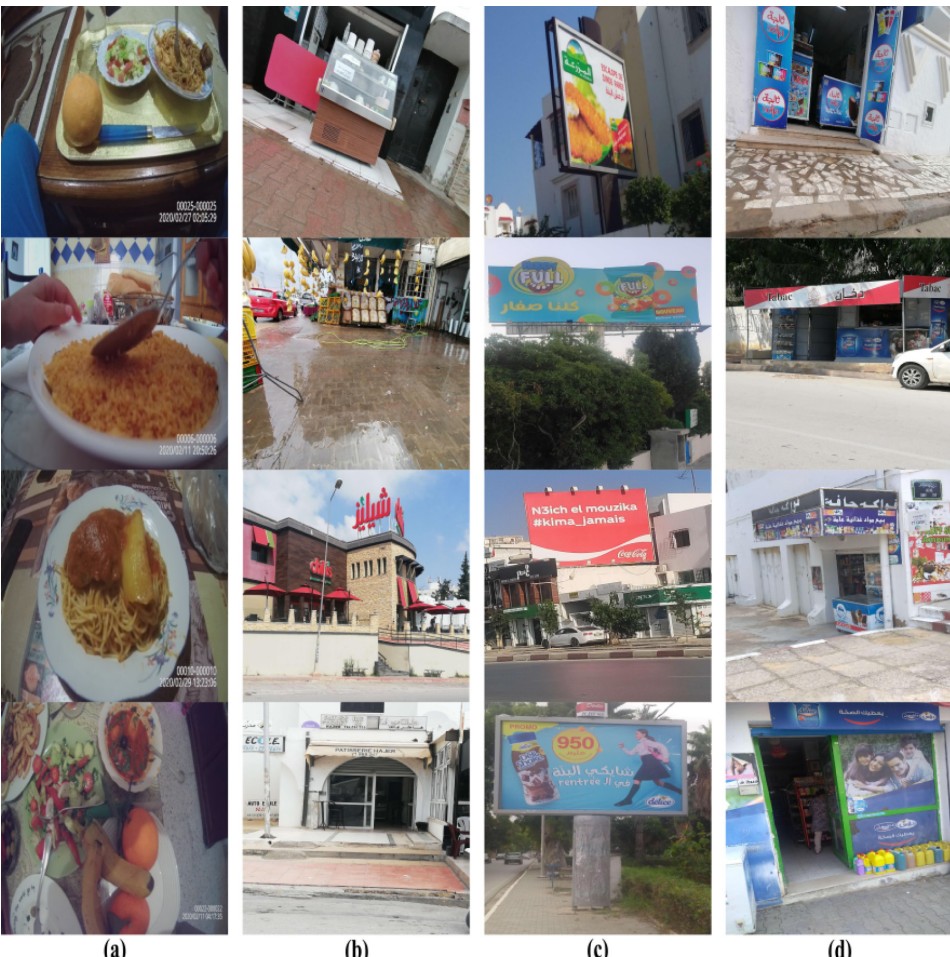

**Fig 2. Example images that contain (a) food items being consumed, (b) food outlets, (c) food advertisements, and (d) both food outlets and food advertisements.**

4- A desktop application to pack together these components by extracting images from the wearable cameras' recorded videos, then running the machine learning model and face blurring technique to generate a set of filtered images.

5- A MobileNet V2 [28] machine learning model to classify the food exposure images into subsequent classes, namely food consumption, food advertisement and food outlet. The food consumption category consists of images that contain food items that are being consumed or about to be consumed. The food advertisement category includes any ads that are related to food such as billboards, storefront ads, etc. Finally, the food outlet category includes images that contain a food outlet such as a supermarket, a shop, a restaurant, a kiosk, a cafe, etc. As an image might belong to multiple of these classes at the same time, our machine learning model is a multilabel classification model and it was trained using both public data and data acquired via crowdsourcing. Our machine learning model achieved an *average f-score of 0.96* on test data.

6- A MobileNet V2 machine learning model that aims to further classify the food consumption images into personal food Consumption or others food consumption, or both. An image belongs to the personal food consumption category if it contains food items that

the child wearing the wearable camera is obviously consuming or is about to consume. An image belongs to the others food consumption category if it contains other people consuming food or about to consume food. Our machine learning model is thus a multi-label classification model, and it was trained using data labeled via crowdsourcing. Our food consumption machine learning model achieved an *average F1-score of 0.95* on test data.

Finally, we deployed our machine-learning-based data-collection system in a real-world case study in Tunisia. In this manuscript, we provide the results of an interim study sample consisting of 265 children aged 11-12 years from 29 schools in Greater Tunis, and we report on the accuracy and precision of the developed system.

## 2 Materials and methods

### 2.1 User-centered design study

Based on recommendations to ensure ethical conduct in visual research [29, 30], we assessed the feasibility and acceptability and aimed to develop a set of design recommendations for using a data-collection tool involving technology to capture exposure to food outlets, items and advertisements among school children in Lebanon and Tunisia. For this purpose, we adopted a user-centered design approach to involve school students, their parents, and school staff, in the design of this data-collection tool. We conducted several qualitative interactive design workshops whereby we discussed potential methods to capture children's exposure to food in their immediate environment, their advantages and challenges of these methods, and whether using a technology-based tool can be suitable and acceptable in these contexts [31]. In the following, we give an overview of the study, summarize its findings, and describe the data-collection protocol designed collectively with the participants in the qualitative workshops.

**2.1.1 Workshops and main findings.** Our user-centered design study was based on a purposive sample of primary schools from Greater Beirut and Greater Tunis. To ensure maximum variation in our sample, schools were selected from different socio-economic backgrounds. Two parallel workshops were conducted in each school: the first one with children from grades 5-6 (aged 10-12 years); and the second one with their parents and the schools' directors and staff. Written consent and assent were obtained from all adults and children who participated in the workshops. After conducting 12 workshops in six schools (n = 2 in Greater Beirut and n = 4 in Greater Tunis), data saturation was reached. In total, 40 students, 31 parents and 17 school staff participated in the workshops, which included a range of activities such as discussions, storyboarding, mind-mapping, and brainstorming. The discussions were recorded and transcribed, the materials (e.g., storyboards, design boards) were collected, and data were thematically analyzed.

Commonly reported challenges that we needed to account for in our system were: invading children's and third parties' privacy, distracting the school children during classes, and obtaining biased data. To overcome these challenges, most participants suggested that a passive-image capturing tool for a short period of time—such as a wearable camera with continuous footage—could be a safe and efficient method to capture children's food exposures. This was the instrument of choice because it minimized the challenges that would have been encountered with active image capture, such as children's distraction at schools and on the roads. However, for it to be acceptable and safe to use, the tool must follow certain conditions as suggested by the participants. In particular, the camera:

1- should be worn for a limited period of time and outside the confines of school to avoid distraction,

2- can be turned off at any time or place, for example, when going to the bathroom or if asked by a third party,

3- should be unobtrusive,

4- should be easily and firmly attached to clothing to avoid interfering with the children's activities,

5- should be password protected and with no removable SD card to ensure the security of the recorded images/footage in case of camera loss, and

6- should not have Internet connection nor Bluetooth capabilities to avoid data leakage.

Additionally, to protect anonymity and avoid privacy invasion, most participants proposed a system that automatically removes all non-food related images from the captured images, followed by parental manual screening to remove unwanted images that were not automatically filtered out. They also suggested automated blurring of faces in all captured footage.

**2.1.2 Data-collection protocol.**   Based on the ethical frameworks and the findings of the interactive workshops, we propose a study protocol to collect data on children's food environment during a typical school day. Our protocol concerns children aged 11 to 12 years as our findings and the literature show that this is the youngest age group capable of appropriately using wearable cameras [32]. The protocol involves a machine-learning-based tool, which includes:

1- a wearable camera that continuously records videos/images of the child's environment,

2- a customized wearable strap with adjustable size and a pocket in which the camera can be held that makes the camera unobtrusive and practical to wear,

3- a machine learning model that filters recorded images to only retain those related to food/beverages (i.e., "food exposure" images),

4- a face blurring technique to blur faces in all the images retained after the first filtering, and

5- a desktop application that can extract images from the wearable cameras' videos, and then run the machine learning model and face blurring technique to generate a set of filtered images.

When conducting research with the proposed tool, ethical approval must be granted from the appropriate ethics committees/institutions. The protocol also specifies that children are only allowed to wear the camera outside the confines of school and have the option to turn the cameras off when needed. Once data are collected, the research team picks up the cameras from the children the following day when children's parents are also invited to the school. During this session, the cameras' footage is transferred to a password-protected computer using a USB cable, in a private room and in the presence of the parents only. The desktop application extracts a frame every 10 seconds from the recorded footage and deletes all the video files once extraction is completed. The rationale behind extracting a frame every 10 seconds is to avoid too many duplicate images capturing the same scene. This application also:

1- deletes all images unrelated to food exposure using a custom machine learning model,

2- blurs all the faces that appear in the retained images using a face blurring technique, and

3- creates for each child (i.e., camera) one folder with the filtered images labeled by the child's ID.

Then, each folder is transferred into one tablet and deleted from the desktop application. Parents are given a tablet that contains the folder specific to their child and are asked to screen and permanently delete any unwanted images. The researchers will not view any recordings and/or images before the end of the two steps of the filtering process. Finally, the filtered set of images gets transferred from the tablets into a password-protected computer maintained by the primary investigator and are then subjected to subsequent machine learning models that classify the retained images into predefined relevant classes such as food consumption, food advertisements and food outlets. The original folders are then permanently deleted from the tablets following the transfer. We adopted this protocol in a real-world case study in Tunisia, described in Section 3.4.

## 2.2 Machine-learning-based data-collection system

Building on the findings of the user-centered design study described in the previous section, we developed a machine-learning-based data-collection system that captures school children's exposure to food in their immediate environment using wearable cameras and machine learning. This system guarantees user privacy and minimizes the risk of invading children's private life. In this section, we first give an overview of the camera model and then describe the machine learning models. Finally, we describe other aspects of the system, such as the face blurring technique and the desktop application used to manage the collected data.

**2.2.1 Camera model.**   To be able to conduct real-world studies to capture food exposure among school children using wearable cameras, the cameras should:

1- be lightweight and small in size to be easily worn by children,

2- be resistant to shock,

3- have long battery life and good storage capacity to record continuous footage for an extended period,

4- have high resolution to ensure good-quality footage,

5- have wide-angle lens to capture as much of the surroundings as possible,

6- have Infrared capabilities to capture footage in low-light or dark settings,

7- have Optical Image Stabilization to reduce motion blur,

8- have a silent-mode option i.e., operating without making any beep, sound, noise, or light to avoid disrupting the children, and

9- have a password-protection option.

The camera's price was also considered to ensure that the system can be used in large-scale studies. We surveyed over 40 different wearable-camera models retrieved from an online search that was conducted at the time of the user-centered design study (Spring 2019). We ordered a sample of the most adequate models (n = 4) to pilot-test them in the field and we opted for the MIUFLY 2K Body Pro model as this camera met all the above specifications. It had an acceptable size and weight, a good resolution, a wide-angle image sensor, a long battery life (9 hours of continuous video recording at 720p), an Infrared sensor, a password-protection option, a silent mode that disables all LED lights and sounds, a vibration motor for operation feedback, a 32 GB built-in flash storage and is resistant to water and shock.

**2.2.2 Food exposure detection model.**   The goal of the food exposure detection model is to automatically classify a given image as a "food exposure" or a "non-food exposure" image. We consider that an image contains food exposure if it displays anything associated with food

such as food items, food markets, food outlets, food ads, restaurants, vending machines, etc. Here, the term food also includes beverages.

**Dataset**. To the best of our knowledge, there are no available datasets that contain images labeled as either containing food exposure or not. For this reason, we opted to build our own dataset. The dataset was automatically constructed and labeled using an open source Web crawler that crawls images from Google and Microsoft Bing search engines [33]. We crawled two classes of images as follows:

1- food exposure class: for this, we used around 290 queries and retrieved at least 2000 images for each query. The queries included "food advertisements", "people eating", the names of the most popular food items available in Lebanon and Tunisia (i.e., settings of our work), the names of the most popular restaurants and food brands in both countries, as well as objects/places that might be related to food and are commonly seen during a typical school day (e.g., cafeteria, kitchen, vending machines, food markets, refrigerators, and open cupboards).

2- Non-food exposure class: for this, we used around 150 queries of objects/places that are not associated with food and are commonly seen during a typical school day (e.g., class-room, playground, interiors of buses, cars, books, bedrooms, living rooms, TV screens, trees, mountains, sea, and animals). As we expected that some of these queries might return food-related images, we sampled 100 images from the results of every non-food exposure query and manually inspected them. We then decided to discard all the queries for which more than 5% of the sampled images contained food-related items. Thus, for each retained query, we retrieved 4,500 images, which is higher than the food exposure class to ensure that our dataset is balanced as we had less queries for the non-food exposure class than the food exposure one.

In total, we crawled more than one million images (n = 1,037,459 images) for around one month from both Google (85%) and Microsoft Bing (15%). This 85-15 split occurred because crawling using Google was faster and returned more images than Microsoft Bing. In addition, the used crawler API provided more filtering and control options when using Google. Each image was labeled as food exposure or not based on the query used to retrieve it.

In this work, we relied on the accuracy of the Web search engines to label our dataset. We acknowledge that relying on the search engines to label the data might result in some incorrect labels. However, since we acquired a very large dataset and we selected the queries to retrieve the images in each class carefully, we hypothesized that the percentage of incorrectly labeled images would be negligible and would not adversely affect the training of the machine learning models. To validate this, we randomly selected four subsets of the dataset, three containing 1000 images each and one 2000 images, and we manually validated the labels of each image in the four samples. The percentage of incorrectly labeled images was 10% on average, as shown in Table 1.

**Table 1. Percentage of noise in four random samples from the dataset.**

|          | Sample Size | Correctly Labeled | Incorrectly Labeled | Noise Percentage |
|----------|-------------|-------------------|---------------------|------------------|
| Sample 1 | 1000        | 907               | 93                  | 9%               |
| Sample 2 | 1000        | 894               | 106                 | 11%              |
| Sample 3 | 1000        | 916               | 84                  | 8%               |
| Sample 4 | 2000        | 1762              | 238                 | 12%              |

Next, we performed some pre-processing and filtering on the over one million crawled images. First, we eliminated duplicate images from the dataset, including those with different resolutions, retaining only 702,096 images. Second, we removed all the images with a resolution lower than 224 x 224, and all the images with transparent backgrounds or watermarks. Third and last, we re-balanced the dataset so that the number of food exposure images was equal to that of the non-food exposure ones by randomly deleting images from the bigger class. Thus, the final training dataset consists of approximately *510,000* images, equally balanced between the two classes.

**Model**. We used the dataset described above to train a deep learning model that can automatically classify whether a given image contains food exposure or not (i.e., a *binary-classification* task). As a first step and as is custom in machine learning projects, we split our dataset as follows: 80% for training (approx. 195,000 images per class), 10% for validation (approx. 25,000 images per class), and 10% for testing (approx. 25,000 images per class). We then trained five different convolutional neural network (CNN) models, using state-of-the-art architectures in deep learning for computer vision, namely VGG16 [34], VGG19 [34], MobileNet V1 [27], and two custom CNNs that we developed from scratch. We then used the validation set to select the best model, the MobileNet V1 model, which we describe in details next.

Our food exposure detection model was a pretrained MobileNet V1 model [27] with weights set based on ImageNet [35]. MobileNet V1 is one of the most heavily-used deep learning models these days. It has achieved superior results compared to many other models for various computer-vision tasks such as the ImageNet competition. Moreover, MobileNet is designed to run efficiently on Mobile phones and thus requires much less computational power compared to other popular models. The whole MobileNet network involves only 4.8 million weights.

To train our model, we loaded MobileNet V1 with its pre-trained ImageNet weights, then the last layer with 1000 classes, corresponding to the ImageNet classes, was modified to represent two classes instead. We then trained this model by updating the weights of all the layers (i.e., training from scratch).

Training the above-described model took place on a local workstation with the following specifications: a Dual Intel Xeon Gold CPU with 28 physical cores and 56 logical cores (threads), a NVIDIA Quadro P2000 GPU with 5 GB of dedicated GPU memory, 64 GB DDR3 RAM, 6 Mbps Network Bandwidth, and 1 TB NVME SSD storage. The total training time on the local workstation took around 2-3 months. We tuned the model's hyperparameters using the validation set and random search. In brief, we used the Adam optimizer with a learning rate raging between 0.0001 and 0.05, and a learning weight decay of 0.0001. The dropout rate was 0.5 and the batch size was set to 32 since the GPU memory of the workstation was not big enough to support higher batch sizes. Data Augmentation was also employed using different techniques such as zooming, tilting, flipping, mirroring and shifting. Finally, we kept on training the model until the training accuracy either bypassed 98.5% or flattened and stopped increasing by more than 0.01 for five successive epochs.

**2.2.3 Food exposure classification model.** The goal of the food exposure model is to classify food-related images captured by the wearable cameras into a hierarchy of food exposure classes. The classes in the food exposure typology are: food consumption, food advertisement and food outlet. The food consumption class consists of images that contain food items that are being consumed or about to be consumed. The food advertisement class includes any ads that are related to food such as billboards, storefront ads, etc. Finally, the food outlet class includes images that contain a food outlet such as a supermarket, a shop, a restaurant, a kiosk, a cafe, etc.

**Table 2. Distribution of images over classes in the Children Trajectory dataset.**

| Class | Count |
|---|---|
| Personal Food Consumption | 1600 |
| Others Food Consumption | 340 |
| Personal and Others Food Consumption | 940 |
| Food Outlet | 380 |
| Food Outlet and Advertisement | 70 |

**Dataset**. We used three different datasets to train our food exposure model. The first dataset, which we refer to as the Children Trajectory dataset consists of food-related images captured using wearable cameras of 265 children from 29 schools in Tunisia. This dataset includes 3,560 food-related images and we used the Labelbox crowdsourcing platform [36] to manually classify the images in the dataset into one or more of our three relevant classes, namely food consumption, food advertisement and food outlet.

To this end, we trained a team of five annotators, who worked at Labelbox as full-time labelers at the time of annotation, on our annotation task. The task was that for each image, they should select one or more of the following categories: personal food consumption, others food consumption, food outlet, and food advertisement. The segregation of food consumption into personal and others was done since our next machine learning model aims to classify food-consumption images into these two subclasses. To ensure high-quality annotations, we reviewed the labeled images through a voting system, where the incorrect labels were given down votes and then they were corrected by the annotators. Moreover, there was a direct communication with the annotators through a shared document where they can ask about ambiguous images. Table 2 shows the distribution of the annotated images over the different classes.

Since the number of images that belong to either the food outlet or the food advertisement classes in the Children Trajectory dataset was significantly low, we used a second dataset to train our food exposure model. This dataset consists of images of food outlets or food advertisements that are located in the neighborhoods of the children's schools (within a range of 800-meter), and that were captured using trained data collectors. We refer to this dataset as the Neighborhood Mapping dataset. Table 3 shows the number of images that belong to each of the two classes (advertisements or outlets).

Finally, we also used the EgocentricFood dataset [37], which is a publicly-available dataset consisting of 5,038 food-related images that were taken using wearable cameras. From this dataset, we sampled 3000 food-consumption images, and 200 food-outlet images. Since the number of images that belong to the food advertisement class was still very small in all datasets, we additionally crawled food advertisement images using a major Web search engine (Google). Table 4 shows the number of images that belong to the different classes in each dataset.

**Model**. We combined all the datasets described above and used them to train a deep learning model to classify food exposure images into *one or more* of the following classes: food consumption (both personal or other), food outlet, and food advertisement. To ensure

**Table 3. Neighborhood Mapping dataset.**

| Class | Count |
|---|---|
| Food Outlet | 2048 |
| Food Advertisement | 25 |
| Food Outlet and Advertisement | 2130 |

**Table 4. Distribution of images in all datasets over the different classes.**

| Dataset | Food Consumption | Food Outlet | Food Outlet + Ads | Food Ads |
|---|---|---|---|---|
| Children Trajectory | 2880 | 380 | 70 | - |
| Neighborhood Mapping | - | 2048 | 2130 | 25 |
| EgocentricFood | 3000 | 160 | - | - |
| Crawled Ads | - | - | - | 512 |
| Total | 5880 | 2588 | 2200 | 537 |

consistency among the images in the combined datasets, we resized all the images to $224 \times 244$. We then split our dataset into 80% for training (8,965 images), 10% for validation (1,120 images) and 10% for testing (1,120 images), in a balanced way among the classes. We then trained three state-of-the-art CNN models, namely MobileNet V1 [27], MobileNet V2 [28], and VGG16 [34] and used the validation set to select the best model, the MobileNet V2 model, which we describe in details next.

We loaded the MobileNet V2 model with pretrained ImageNet [35] weights. We then replaced the output layer of the pretrained MobileNet V2 model with a GlobalAveragePooling2D layer, followed by a dense layer consisting of 256 neurons and a dropout regularization layer (twice). Finally, we also added one last layer, which is a dense layer, consisting of three neurons, each with a Sigmoid activation function that outputs independent probabilities for each one of our three classes. Finally, to fine tune the model using our combined dataset, we unfroze the last 56 layers of our modified MobileNet V2 model and only trained those. We then tuned the hyperparameters of the model using the validation set and random search. Our best performing model was trained for 20 epochs using the Adam optimizer, with a batch size of 64, and a learning rate of 0.001. We then continued to train that model for 15 more epochs with a learning rate to 0.0001. The model was trained on the same local workstation that we used to train our first model, the food exposure detection model.

**2.2.4 Food consumption classification model.** The goal of the food consumption model is to classify food-consumption images into two subsequent subclasses, namely personal food consumption and others food consumption. The personal food consumption class consists of images in which the child wearing the camera is consuming or about to consume food, whereas the others food consumption class consists of images in which other people are consuming or about to consume food. Note that an image can belong to both the personal food consumption and the others food consumption categories, and hence this is a multi-label classification task.

**Dataset**. We constructed the dataset to train our food consumption model by extracting the food consumption images from of the Children Trajectory dataset that was used to train our food exposure classification model. This distribution of the food consumption images over the two subclasses is shown in Table 5. Fig 3 shows sample images for (a) personal food consumption, (b) others food consumption, and (c) personal and others food consumption.

**Table 5. Distribution of food consumption images over the two subclasses in the Children Trajectory dataset.**

| Class | Count |
|---|---|
| Personal Food Consumption | 1600 |
| Others Food Consumption | 340 |
| Personal and Others Food Consumption | 940 |

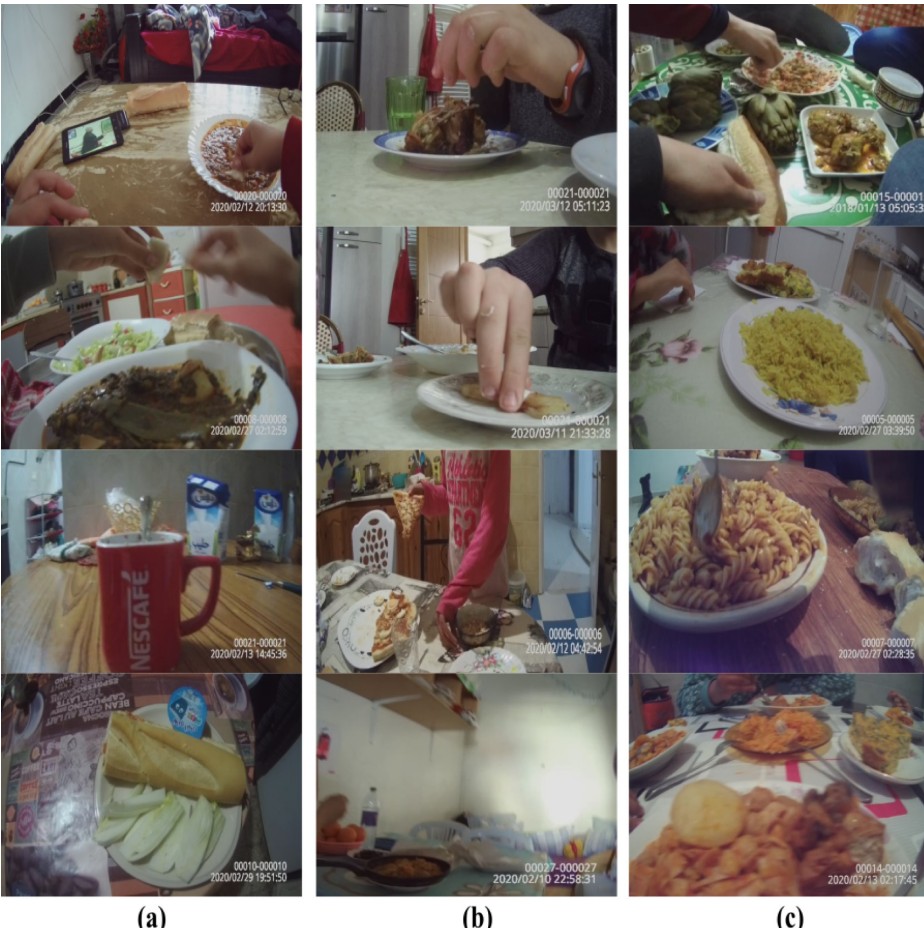

**Fig 3. Example food consumption images from the Children Trajectory dataset.** (a) corresponds to personal food consumption, (b) corresponds to others food consumption, and (c) corresponds to both personal and others food consumption.

**Model**. To train our food consumption model using the subset of the Children Trajectory dataset corresponding to food consumption, we split the dataset into 70% for training (2,016 images), 10% for validation (297) and 20% for testing (576) in a balanced way among the two classes. Our food consumption model was based on a pretrained MobileNet V2 model that has the same architecture as the food exposure model described above, but instead of using three neurons in the output layer, we used only two since we this model aims to classify images into two classes (personal food consumption and others food consumption). The hyperparameters of the model were then tuned using the validation set and random search. The model was trained for 25 epochs, with a learning rate of 0.001, Adam optimizer, and a batch size of 32, on the same local workstation used to train our first two machine learning models.

**2.2.5 Face blurring technique.** A critical feature of our system is to blur all faces in any captured images automatically. To select the most suitable face blurring technique, we tested three different approaches: (1) the OPENCV's library [38]; (2) an open source CNN named MTCNN [39], which was reported to have an accuracy of up to 90% [39]; and (3) a python

library called CVLIB, which performs face detection [40]. The first two approaches detect faces and draw a virtual bounding box around them.

We ran the three techniques, separately, on two sampled sets of images retrieved from our first dataset crawled from the Web (2000 images each). The OPENCV's library did not perform well and tended to blur objects instead of human faces. The MCTNN was able to detect and blur 88.91% and 91.53% of faces in the first and second samples, respectively. Only cropped faces or faces in the far background were usually not detected and hence not blurred. The CVLIB library yielded almost the same results as the MTCNN. Thus, we decided to use the MTCNN followed by the CVLIB library to maximize face-detection accuracy for our machine-learning-based data-collection system. To blur faces, we take the x-y coordinates of the virtual bounding boxes that contain faces and apply Gaussian Blurring to the bounding box area. The accuracy of face blurring, when both methods were used in a pipeline, was around 91% on average.

**2.2.6 Desktop application.** To retrieve the data stored on the wearable cameras, we developed a desktop application that uses the food exposure detection model (i.e, our first machine learning model) and the face blurring technique described above. The front-end of this application was written in Java, while the back-end used Python. The application was designed to retrieve data from a maximum of 10 cameras simultaneously. Using this application, the research team is able to (1) connect the cameras (on which the data is collected) to the computer through USB ports, and (2) enter the passwords of the cameras to enable USB access. The application then checks if all the cameras have footage and notifies the team of any empty cameras. Then, the research team members enter the school ID, child ID and camera ID onto the application to initiate data transfer.

Since the cameras record continuous videos, the application first extracts a frame every 10 seconds and deletes all the video files once extraction is completed. It then employs the fist machine learning model to identify and retain all food exposure images in the collected data and discard any other footage. Next, the application uses the face blurring method to blur all the faces in the retained images. After the images are exported, filtered, and faces in them are blurred, the application generates a brief report of the total number and percentage of retained food exposure images. In case of any error during the process, a warning message is generated at the end describing the error and how it was handled. Our machine-learning-based data-collection tool was thoroughly designed so that no data loss would occur in case of any unexpected failure caused by a system or a hardware error at any point during the process while guaranteeing data protection throughout the entire process.

# 3 Results and discussion

## 3.1 Food exposure detection model

We tested our MobileNet V1 food exposure detection model described in Section 2.2.2 on the test data (i.e., the subset of its dataset consisting of around 25,000 images not used in training nor validation). We obtained a test accuracy of *92.53%* and a test F1-score of *0.9204*. Table 6 shows the test performance of the model using various metrics. Our trained model and the dataset used to train it are publicly available [41].

**Table 6. Performance of the MobileNet V1 food exposure detection model on test data.**

| Metric | Accuracy | Precision | Recall | F1-Score | ROC AUC |
|---|---|---|---|---|---|
| Value | 0.9253 | 0.9541 | 0.8967 | 0.9204 | 0.9504 |

**Table 7. Performance of the MobileNet V2 food exposure model on test data.**

| Class | Precision | Recall | F1-score |
|---|---|---|---|
| Food Consumption | 0.99 | 0.99 | 0.99 |
| Food Outlet | 0.98 | 0.99 | 0.98 |
| Food Advertisement | 0.95 | 0.93 | 0.93 |
| Average | 0.97 | 0.97 | 0.96 |

### 3.2 Food exposure classification model

Table 7 show the results of our food exposure model described in Section 2.2.3 on its test data. As can be seen from the table, the model achieved very high results in terms of precision, recall and F1-score for all three classes. The food advertisement class had the lowest precision and recall since some of the test images that belong to that class contain food advertisements that are very small and thus barely visible in the images.

In addition, we used LIME [42], which stands for Local Interpretable Model-agnostic Explanation, to explain the model's decisions on sample images by extracting the regions that are responsible for the classifier's predictions. As can be seen in Fig 4, the model was able to correctly classify image (a) as food consumption by focusing on the region of pixels that contain food items. Similarly, the model was able to correctly classify image (b) as food outlet and advertisement by focusing on the region of pixels that corresponds to an outlet, and the region of pixels that corresponds to an advertisement.

### 3.3 Food consumption classification model

The results of our food consumption model described in Section 2.2.4 on its test data are shown in Table 8. As can be seen from the table, the model achieved very high results in terms

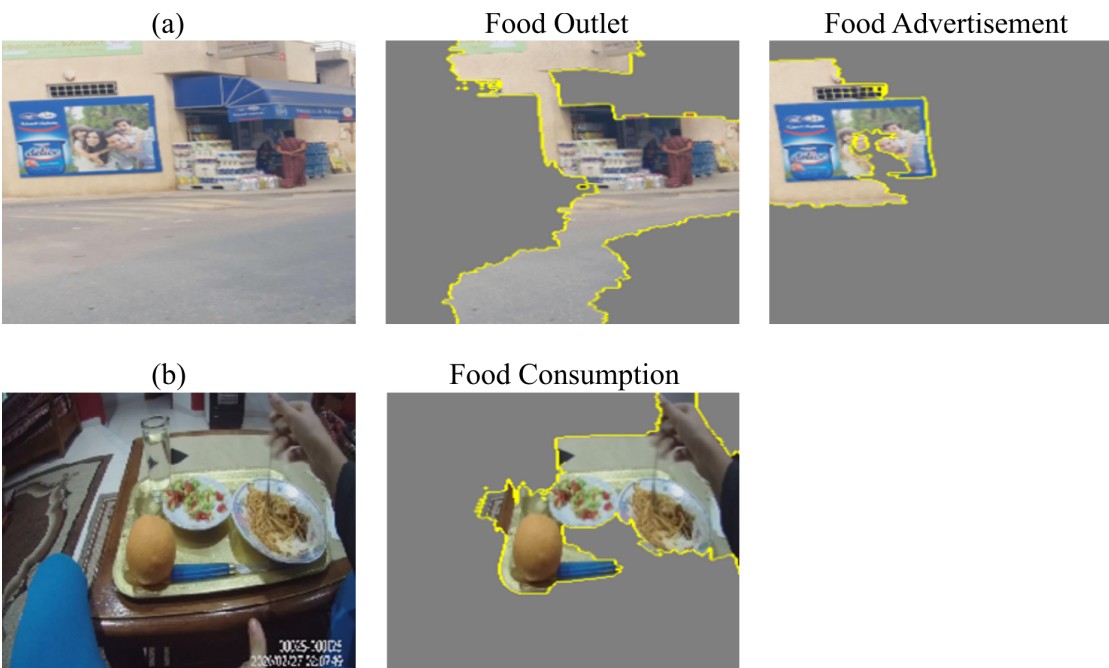

**Fig 4. LIME results on sample images for the food exposure model.**

**Table 8. Performance of the MobileNet V2 food consumption model on test data.**

| Class | Precision | Recall | F1-score |
|---|---|---|---|
| Personal Food Consumption | 0.97 | 0.99 | 0.98 |
| Others Food Consumption | 0.97 | 0.89 | 0.93 |
| Average | 0.97 | 0.94 | 0.95 |

of precision, recall and F1-score for the personal food consumption class. However, the others food consumption model had a slightly lower recall since some of the images contained other people consuming food at the same table as the child wearing the camera, however, the camera was directed towards the table and thus the captured image did not clearly show the other people.

Similar to the previous model, we also used LIME to explain the model's output on a sample image, which is shown in Fig 5. The sample image was classified as personal and others food consumption because there is a dish that is directly in front the camera wearer on one side, and there is another person who is consuming food at the same table on the other side.

### 3.4 Case study

Our machine-learning-based data-collection system described in the previous section was deployed in a case study in Tunisia and is currently being deployed in Lebanon [43]. In this section, we present the results from the cross-sectional study that was conducted only in Tunisia.

**3.4.1 Setting.** Our case study involved a representative sample of 8-11 years-old school children (grades 4, 5 and 6) registered in schools in the Greater Tunis. A stratified two-stage sampling was used: first, a random sample of 50 schools was selected; then a random sample of 50 children were recruited from each school. Finally, we took a random subsample of around 10 children aged 11-12 years (grades 5 and 6) from each of the 50 schools. This study was approved by the relevant Ethics Committee at the National Institute of Nutritional and Food Technology (INNTA) in Tunisia.

We adopted the protocol described in Section 2.1 to collect data using wearable cameras, including obtaining oral consent from schools' directors, as well as parents and children. Field-work took place from January to March 2020 and then from October to November 2020 as Tunisian schools had to close in between in response to the emergent COVID-19 pandemic. We present here the results of the interim study sample that was recruited before the COVID-19 lockdown. In total, 265 children aged 11 to12 years old were recruited from 29 schools in Greater Tunis. Most schools (86%) had 8 or more participants.

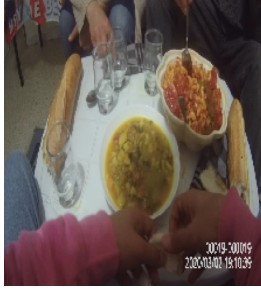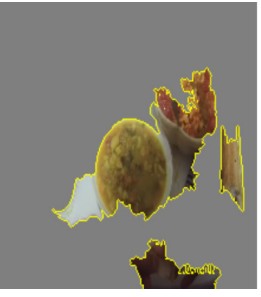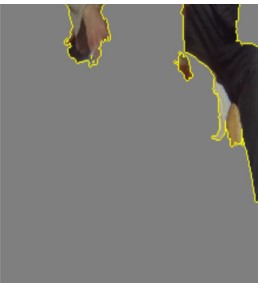

**Fig 5. LIME results on a sample image for the food consumption model.**

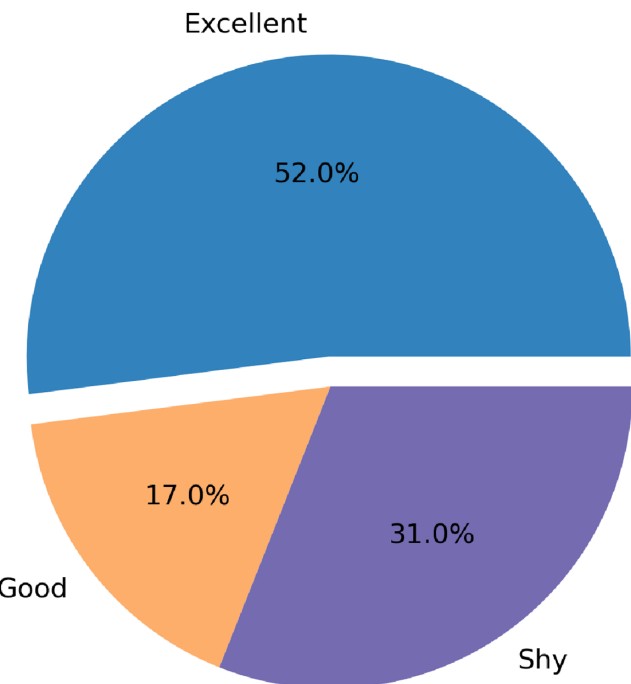

**Fig 6. School participation rate.**

**3.4.2 Results.** The throughput of our food exposure detection model was around 1,100 images per minute and the throughput of the face blurring module was close to 600 images every 4 minutes (around 150 images per minute). Overall, we managed to collect 788.26 hours (695.29 GB) of footage time and 567,506 images were extracted from this footage. We then ran our machine learning model and ended up with a total of 61,857 images (6.92 GB) related to food exposure (all the rest were deleted). Despite parents being invited to filter out any retained images, none of them attended the data review and manual filtering sessions.

To assess the feasibility of our machine-learning-based data-collection system, we computed the school participation rate among our interim study sample. The school participation rate was calculated as the number of empty cameras returned by participants from each school, over the total number of participants in this school. Participation was considered excellent if the ratio ranges between 0 and 0.25, good if it ranges between 0.25 and 0.5, and shy if it ranges between 0.5 and 1. As can be seen in Fig 6, the majority of schools had an excellent participation (52%).

We further computed the participation rate at the child's level and we show the results in Fig 7. Over 53% of participants had excellent (i.e., more than 4 hours of footage recorded) or good (between 1 and 4 hours of footage) participation rate and only around 8% did not participate at all in the study (i.e., had empty cameras). We received empty cameras as some children forgot to turn their cameras on and others changed their minds and decided not to wear them anymore.

Finally, to validate the effectiveness of the machine learning model in detecting food exposure images, we took a random sample of 25 children and manually inspected the corresponding final sets of retained images. The size of each set of images ranged between 100 and 500 depending on the amount of recorded footage of each child. Since parents did not filter any images in our case study, we were able to verify the model's precision by identifying false-

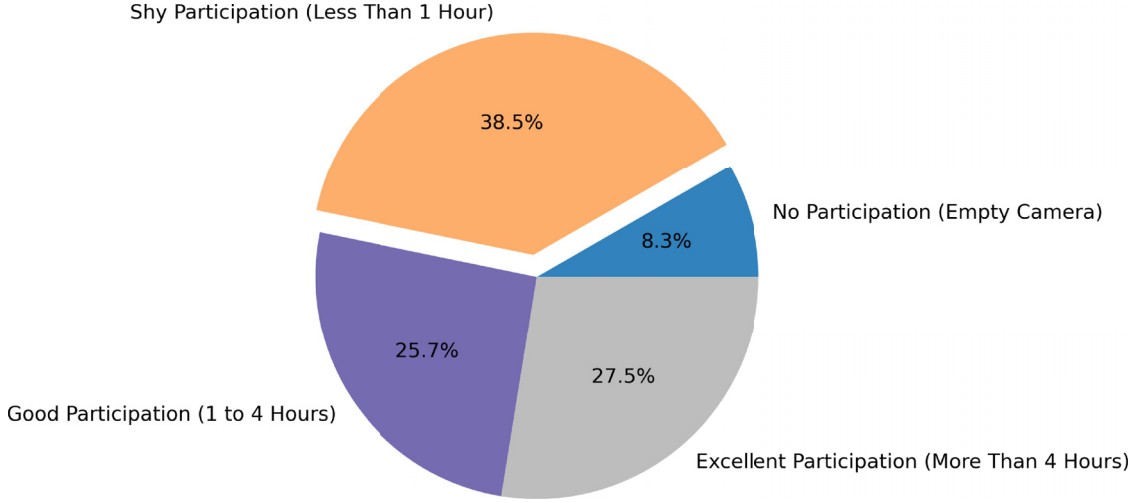

**Fig 7. Children participation rate.**

positive instances (i.e., images that were not actually related to food exposure but were retained). Since all other images that our machine learning model did not classify as food exposure were immediately deleted, we could not measure the model's recall (i.e., false-negative rate).

The model precision ranged between 82.23% and 93.57% depending on the sample (i.e., the false-positive rate ranged between 6.43% and 17.77%). Our machine learning model was thus able to generalize well on real data despite the fact that the model was trained on data collected from the Web. Upon analyzing misclassified images, we observed that many of them were blurry due to fast motion or because they were taken in low-light settings. Fig 8 shows a sample of the food exposure images that were retained by our data-collection tool and in which all the faces were automatically blurred.

Next, we applied the food exposure and the food consumption classification models on the food exposure images detected by the first machine learning model as described above. The food exposure model classified the vast majority of the food exposure images as food consumption (95%) and only 4% as food outlets. Among the food consumption images, 69% were classified by the food consumption model as personal food consumption, only 8% were classified as others food consumption, and the remaining 23% were classified as both personal and others food consumption.

## 3.5 Summary and outlook

In this manuscript, we described an end-to-end machine-learning-based system to capture food exposure among school children using wearable cameras. Our system has several technical features to ensure that data collection is performed ethically, securely, and in a scalable manner. The development of the system's features was guided by a user-centered design study on the acceptability and feasibility of using wearable cameras among children. Our system employs three different deep learning models, which are able to automatically detect images related to food exposure and to classify those into fine-grained classes. These classes are images that contain food items consumed by the child wearing the camera or others, images that contain food advertisements and images that contain food outlets. To train the machine learning

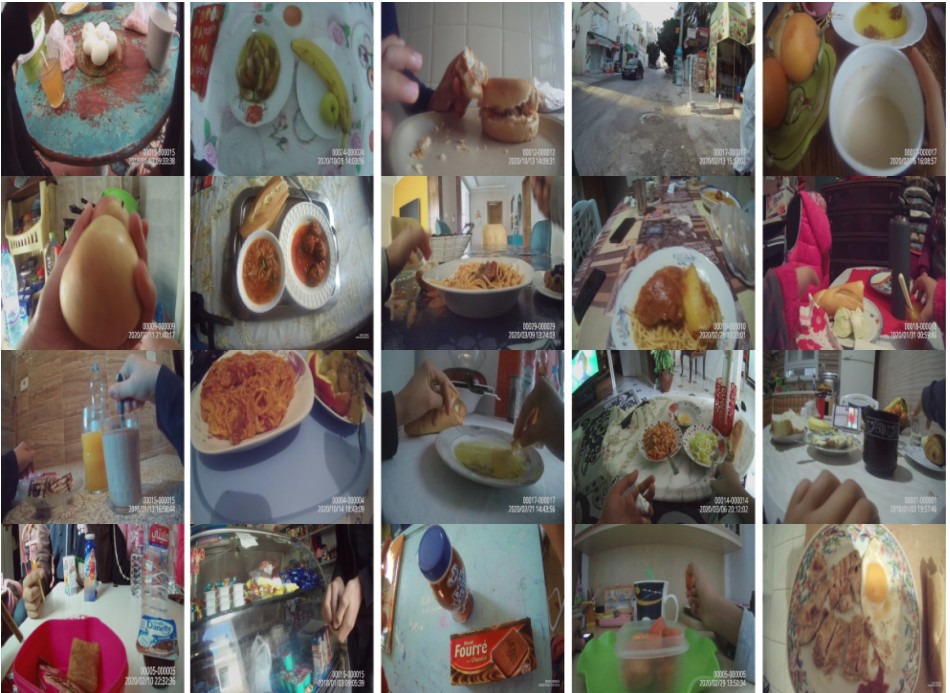

**Fig 8. Some example food exposure images successfully captured by our system.**

models used by our system, we constructed various large datasets of labeled images about food exposure, by crawling images from the Web and labeling images captured through wearable cameras using crowdsourcing. We deployed our system in a real case study and provided some analysis of the data collected in that study.

We believe that our work paves the way for many more studies of this nature and provides many useful lessons related to the issue of using wearable technologies to document food experiences, particularly among children. We plan to use our trained models in another user study in Greater Beirut, Lebanon. Since our models detect whether an image contains food exposure or not, and for those that contain food exposure, what type of exposure it is, we believe our models do not require any further training and can be deployed successfully in any food context. In the future, we plan to train further machine learning models to automatically classify food consumption images based on their healthiness into various groups using the NOVA food classification system. The filtered and labeled images and metrics relative to each child will then be analyzed along with other individual-level, school-level and community-level variables collected to inform school and community-level policies that foster a healthier food environment for school children in Greater Beirut and Greater Tunis.

## Acknowledgments

We would like to thank the "School and community drivers of child diets in Arab cities; identifying levers for intervention" SCALE Research Group for supporting this work (Akik Chaza, Chalak Ali, Doggui Radhouene, El-Helou Nehmat, Jamaluddine Zeina, Khammassi Marwa, Safadi Gloria, Sassi Sonia, Skhiri Hajer, Traissac Pierre, and Sarah Katerji).

## Author Contributions

**Conceptualization:** Shady Elbassuoni, Hala Ghattas, Jalila El Ati, Aline Semaan, Christelle Akl, Reem Talhouk.

**Data curation:** Yorgo Zoughby, Aline Semaan, Christelle Akl, Tarek Trabelsi, Houda Ben Gharbia, Zoulfikar Shmayssani.

**Formal analysis:** Aline Semaan.

**Funding acquisition:** Hala Ghattas, Jalila El Ati.

**Methodology:** Shady Elbassuoni, Hala Ghattas, Jalila El Ati, Yorgo Zoughby, Aline Semaan, Christelle Akl, Reem Talhouk, Zoulfikar Shmayssani.

**Project administration:** Hala Ghattas, Jalila El Ati.

**Software:** Yorgo Zoughby, Zoulfikar Shmayssani.

**Supervision:** Shady Elbassuoni, Hala Ghattas, Jalila El Ati.

**Validation:** Aline Semaan, Christelle Akl, Tarek Trabelsi, Reem Talhouk, Houda Ben Gharbia.

**Writing – original draft:** Shady Elbassuoni, Yorgo Zoughby, Zoulfikar Shmayssani.

**Writing – review & editing:** Shady Elbassuoni, Hala Ghattas, Jalila El Ati, Aline Semaan, Christelle Akl, Tarek Trabelsi, Reem Talhouk, Aya Mourad.

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
