## [Decision Letter · Decision Letter 0]

2 Dec 2022

PDIG-D-22-00308

Capturing Children Food Exposure Using Wearable Cameras and Deep Learning

PLOS Digital Health

Dear Dr. Elbassuoni,

Thank you for submitting your manuscript to PLOS Digital Health. After careful consideration, we feel that it has merit but does not fully meet PLOS Digital Health's publication criteria as it currently stands. Therefore, we invite you to submit a revised version of the manuscript that addresses the points raised during the review process.

All reviewers generally agreed that the topic of this paper is interesting and within the scope of PLOS Digital Health. However, some key aspects of the paper such as model training, model generalization and image copyrights would need to be fully addressed in the revision. Details of reviewers' comments can be found below.

Please submit your revised manuscript within 60 days Jan 31 2023 11:59PM. If you will need more time than this to complete your revisions, please reply to this message or contact the journal office at digitalhealth@plos.org. Please include the following items when submitting your revised manuscript:

We look forward to receiving your revised manuscript.

Kind regards,

Nicole Yee-Key Li-Jessen

Academic Editor

PLOS Digital Health

Journal Requirements:

2. We ask that a manuscript source file is provided at Revision. Please upload your manuscript file as a .doc, .docx, .rtf or .tex.

3. Please provide separate figure files in .tif or .eps format only and remove any figures embedded in your manuscript file. Please also ensure that all files are under our size limit of 10MB.

4. Figure 2, 3, 4 and 8 include an image of an identifiable person. Please provide written confirmation or release forms, signed by the subject(s) (or their guardian), giving permission to be photographed and to have their images published under a Creative Commons license. You may upload permission forms to your submission file inventory as item type 'Other'. Otherwise, we kindly request that you remove the photograph.

5. Please remove any photos of children (Figures 2, 3, 4 and 8) from your submission.

Additional Editor Comments (if provided):

Reviewers' comments:

Reviewer's Responses to Questions

**Comments to the Author**

1. Does this manuscript meet PLOS Digital Health’s publication criteria? Is the manuscript technically sound, and do the data support the conclusions? The manuscript must describe methodologically and ethically rigorous research with conclusions that are appropriately drawn based on the data presented.

Reviewer #1: Yes

Reviewer #2: Yes

Reviewer #3: Yes

2. Has the statistical analysis been performed appropriately and rigorously?

Reviewer #1: Yes

Reviewer #2: Yes

Reviewer #3: Yes

3. Have the authors made all data underlying the findings in their manuscript fully available (please refer to the Data Availability Statement at the start of the manuscript PDF file)?

Reviewer #1: No

Reviewer #2: Yes

Reviewer #3: Yes

4. Is the manuscript presented in an intelligible fashion and written in standard English?

Reviewer #1: Yes

Reviewer #2: Yes

Reviewer #3: Yes

5. Review Comments to the Author

Reviewer #1: The authors have presented a deep and broad study on the use of deep learning and convolutional neural networks for meassuring food exposure on children. This paper includes data colletion, model training and evaluation and all the workflow surrounding those steps.

The paper is interesting and the authors have made quite and effort for developing the work. In my opinion, there are some aspects that should be clarified and improved in order for considering it for being accepted:

- The authors use both MobileNet V1 and V2. What is the reason for that? Have you tried using either one of the other for all the experiments in order to justify why different models are used throughout the paper?

- Why are the models using pre-trained weights from ImageNet if they are being trained from scratch?

- Why is a multilabel model needed? To me it seems that a binary clasification model could have been used. Moreover, if you train a multilabel system, why is a two-step cascaded system used? A single multilabel model could have been trained including food exposure and food consumption.

- The paper has too many lists, which, in my opinion, does not help with the readability of the paper and confuses a bit.

- Retrieving 2000 images for each query in the food exposure experiment seems to be a lot to me. Have you tried using other configurations for acquiring the images?

- The authors mention a few times in the paper that classes should be balanced and that some images are removed in order to have this aforementioned balance in the datasets. Why is this needed? If you're using TensorFlow you could use the class_weight parameter to benefit from more images while not having balanced classes.

- There is too much detail given for some specific things that I would recommend removing. Particularly, those lines in which the evolution of MobileNet from V1 to V2 is explained, as well as those lines related to how much disk space is used when installing libraries or with the datasets used.

- What would happen in the case that these models are used in a completely different country in terms of food culture? The models seems to be trained on a specific local food variety. I wonder how this would work on countries like Japan, USA, Mexico, etc. Wouldn't it be better to train the model on a dataset with a higher heterogeneity?

Reviewer #2: Line 72: Missing “(a)” for food consumption image in Figure 2

Line 226: “desktop application extracts a frame every 10 seconds” What is the rationale behind this time frame, do similar tools follow this frame? It would be useful to clarify this for the reader.

Great example of user-centered research to inform best practices in developing a tool that is culturally appropriate and acceptable in low-middle income countries like Lebanon and Tunisia. This is a timely study given the explosive increase in obesity figures in the region. Further validation of this tool among other age groups (adolescents and adults) could pave the way for a shift in dietary intake methodology. The existing gold standard for measuring dietary intake data is weighed food records and in some instances 24 hour recalls are used to validate novel approaches such as digital cameras. Are there plans to validate the measurement tool using other dietary assessment methods? Authors mentioned future plans to classify healthy food items using NOVA. It may be worth considering to incorporate volume estimation, and nutritional analysis to further strengthen this as a potential tool to replace conventional dietary assessment methods.

Reviewer #3: General

This is an interesting and useful paper. As wearable cameras become more common, methods to detect and extract information will be increasingly needed by researchers. This paper is a good step forward in that direction. I’m also pleased to see that the code and model are available. I have only one significant concern, which is the way that the test accuracy was measured (see comments under Methods). 

Introduction

I know that the term "AI" has been used elsewhere, but I find it silly and unscientific. Nothing about this model represent 'intelligence'. "Machine learning" is the more accepted term in academia.

Methods

In general I see no problem with training a model using images scraped from the web (there is a slightly thorny issue of copyright that I hope that authors considered, but I don't think it is a scientific issue). However, I am a bit concerned that the accuracy reported throughout the paper is that of the test set derived from the web images. That is not a measure of accuracy that is relevant to the deployment of the system. We have used and coded wearable camera data, and the images tend to be slightly blurry, not have the full item of interest in the frame, have substantial variance in lighting, etc. You have a dataset that was used for training the classification model – why was this not used as the test data for the detection model? And, if it wasn’t, you could compute and report accuracy post hoc?

The above comment also applies to the classification models. The only metric that matters is if the model is accurate in the type of data you want to deploy it to (i.e., free-living wearable camera data). Therefore, only these images should be contained in the test set. I know that this is somewhat tested in a small case study, but it should also have been part of the model evaluation.

6. PLOS authors have the option to publish the peer review history of their article (what does this mean?). If published, this will include your full peer review and any attached files.

**Do you want your identity to be public for this peer review?** For information about this choice, including consent withdrawal, please see our Privacy Policy.

Reviewer #1: Yes: Juan P. Dominguez-Morales

Reviewer #2: No

Reviewer #3: No

---

## [Decision Letter · Decision Letter 1]

9 Feb 2023

Capturing Children Food Exposure Using Wearable Cameras and Deep Learning

PDIG-D-22-00308R1

Dear Dr. Elbassuoni,

We are pleased to inform you that your manuscript 'Capturing Children Food Exposure Using Wearable Cameras and Deep Learning' has been provisionally accepted for publication in PLOS Digital Health.

Best regards,

Nicole Yee-Key Li-Jessen

Academic Editor

PLOS Digital Health

Reviewer Comments (if any, and for reference):

Reviewer's Responses to Questions

**Comments to the Author**

1. If the authors have adequately addressed your comments raised in a previous round of review and you feel that this manuscript is now acceptable for publication, you may indicate that here to bypass the “Comments to the Author” section, enter your conflict of interest statement in the “Confidential to Editor” section, and submit your "Accept" recommendation.

Reviewer #2: All comments have been addressed

Reviewer #3: All comments have been addressed

2. Does this manuscript meet PLOS Digital Health’s publication criteria? Is the manuscript technically sound, and do the data support the conclusions? The manuscript must describe methodologically and ethically rigorous research with conclusions that are appropriately drawn based on the data presented.

Reviewer #2: (No Response)

Reviewer #3: Yes

3. Has the statistical analysis been performed appropriately and rigorously?

Reviewer #2: (No Response)

Reviewer #3: Yes

4. Have the authors made all data underlying the findings in their manuscript fully available (please refer to the Data Availability Statement at the start of the manuscript PDF file)?

Reviewer #2: (No Response)

Reviewer #3: Yes

5. Is the manuscript presented in an intelligible fashion and written in standard English?

Reviewer #2: (No Response)

Reviewer #3: Yes

6. Review Comments to the Author

Reviewer #2: (No Response)

Reviewer #3: (No Response)

7. PLOS authors have the option to publish the peer review history of their article (what does this mean?). If published, this will include your full peer review and any attached files.

**Do you want your identity to be public for this peer review?** For information about this choice, including consent withdrawal, please see our Privacy Policy.

Reviewer #2: No

Reviewer #3: No
